# *Calotropis gigantea* stem bark extract induced apoptosis related to ROS and ATP production in colon cancer cells

Thanwarat Winitchaikul[1], Suphunwadee Sawong[1], Damratsamon Surangkul[2], Metawee Srikummool[2], Julintorn Somran[3], Dumrongsak Pekthong[4], Kittiya Kamonlakorn[5], Pranee Nangngam[6], Supawadee Parhira[7]☯*, Piyarat Srisawang[1]☯*

1 Faculty of Medical Science, Department of Physiology, Naresuan University, Phitsanulok, Thailand, 2 Faculty of Medical Science, Department of Biochemistry, Naresuan University, Phitsanulok, Thailand, 3 Faculty of Medicine, Department of Pathology, Naresuan University, Phitsanulok, Thailand, 4 Faculty of Pharmaceutical Sciences, Department of Pharmacy Practice, Naresuan University, Phitsanulok, Thailand, 5 Faculty of Pharmaceutical Sciences, Department of Pharmaceutical Chemistry and Pharmacognosy, Naresuan University, Phitsanulok, Thailand, 6 Faculty of Science, Department of Biology, Naresuan University, Phitsanulok, Thailand, 7 Faculty of Pharmaceutical Sciences, Department of Pharmaceutical Technology, Naresuan University, Phitsanulok, Thailand

☯ These authors contributed equally to this work.
* supawadeep@nu.ac.th (SP); piyarats@nu.ac.th (PS)

**Data Availability Statement:** All relevant data are within the manuscript.

## Abstract

Conventional chemotherapeutic agents for colorectal cancer (CRC) cause systemic side effects and eventually become less efficacious owing to the development of drug resistance in cancer cells. Therefore, new therapeutic regimens have focused on the use of natural products. The anticancer activity of several parts of *Calotropis gigantea* has been reported; however, the effects of its stem bark extract on inhibition of cancer cell proliferation have not yet been examined. In this study, the anticancer activity of *C. gigantea* stem bark extract, both alone and in combination with 5-fluorouracil (5-FU), was evaluated. A crude ethanolic extract was prepared from dry, powdered *C. gigantea* barks using 95% ethanol. This was then partitioned to obtain dichloromethane (CGDCM), ethyl acetate, and water fractions. Quantitative analysis of the constituent secondary metabolites and calotropin was performed. These fractions exhibited cytotoxicity in HCT116 and HT-29 cells, with CGDCM showing the highest potency in both the cell lines. A combination of CGDCM and 5-FU significantly enhanced the cytotoxic effect. Moreover, the resistance of normal fibroblast, HFF-1, cells to this combination demonstrated its safety in normal cells. The combination significantly enhanced apoptosis through the mitochondria-dependent pathway. Additionally, the combination reduced adenosine triphosphate production and increased the production of reactive oxygen species, demonstrating the mechanisms involved in the induction of apoptosis. Our results suggest that CGDCM is a promising anti-cancer agent and may enhance apoptosis induction by 5-FU in the treatment of CRC, while minimizing toxicity toward healthy cells.

**Funding:** SP and PS received grant supported from Thailand Science Research and Innovation, [Grant NO. R2564B007]. TW received the graduate thesis funding from the Department of Physiology, Faculty of Medical Science, Naresuan University, Phitsanulok, Thailand [Grant NO. 61063417]. The funders had no role in study design, data collection and analysis, decision to publish, or preparation of the manuscript.

# Introduction

Cancers are among the most serious health problems globally. The World Health Organization (WHO) reported that cancers were responsible for approximately 30% of all premature deaths from non-chronic diseases in adults aged 30–69 in 2020. In 2018, approximately 18.1 million people worldwide had cancer, and approximately 9.6 million died of the disease. The WHO estimates that by 2040, this number is likely to almost double. Lung cancer is the most frequently diagnosed type of cancer, followed very closely by breast cancer in females (11.6% of all cases), and colorectal cancers (CRC) (10.2%). Lung cancer is also responsible for the largest proportion of cancer-related deaths (18.4%), followed by CRC (9.2%) and stomach cancer (8.2%) [1]. This emphasizes the severity of colon cancer both in terms of the global prevalence and mortality of the disease. Although, standard chemotherapeutic agents, such as irinotecan, oxaliplatin, capecitabine, and 5-fluorouracil (5-FU), are thought to be effective in the treatment of CRC, a high incidence of drug resistance, recurrence, and side effects has been reported [2]. Adverse effects associated with the use of 5-FU include mucositis, diarrhea, gastrointestinal toxicity, and myelotoxicity [3]. Therefore, guidelines for minimizing the limitations and improving therapeutic efficacy in CRC have focused on the use of combination regimens of chemotherapeutic agents, along with compounds derived from natural products.

*Calotropis gigantea* (Apocynaceae, Asclepedaceae) is a plant widely grown in Africa, Eastern Asia, and Southeast Asia, including Thailand. This plant is rich in cardenolides [4–6] and other secondary metabolites, including pregnanones [7], triterpenes [8], and triterpenoids [9]. The extracts and secondary metabolites isolated from several parts of this plant have been reported to exhibit a range of pharmacological activities, including anti-inflammatory [10] and anticancer effects [4,6]. Compounds isolated from the milkweed plants (family: Apocynaceae or Asclepedaceae), such as 12,16-dihydroxycalotropin, calotropin, corotoxigenin 3-O-glycoside, desglucouzarin, uscharin, and 2″-oxovoruscharin, have been explored as potential anticancer agents, owing to their ability to induce cell death through apoptosis and other pathways [11–13]. Recently, an ethanolic extract of the aerial parts of *C. gigantea* was found to induce apoptosis in A549 and NCI-H1299 non-small cell lung carcinoma cells through the activation of extrinsic and intrinsic pathways, cell cycle arrest, and generation of reactive oxygen species (ROS) [14]. The extracts of *C. gigantea* flowers, roots, root bark, and leaves were shown to exhibit cytotoxicity in *in vitro* and *in vivo* models [15–17]. Furthermore, cardenolides isolated from the bark of *C. gigantea* were able to inhibit the proliferation of HeLa and A549 cells [18]. *C. gigantea* has also been shown to augment the responses of cancer cells to other forms of cancer therapy. Coroglaucigenin, a cardenolide isolated from the stem and leaves of *C. gigantea*, enhances the sensitivity of human lung cancer cells to radiation therapy [19]. However, the mechanism underlying the induction of apoptosis in cancer cells by the stem bark extract of *C. gigantea* has not yet been evaluated.

The standard chemotherapy for the treatment of CRC contains 5-FU as an adjuvant and neoadjuvant, to achieve positive clinical outcomes [3,20,21]. Plant extracts have been reported to improve the cytotoxic effect of 5-FU in cancer cells. *Piper betle* was found to enhance the anticancer activity of 5-FU in HT-29 and HCT116 colon cancer cells [22]. A longan flower extract was found to produce synergistic anticancer effects in CRC cells when used in combination with 5-FU through mitochondria-dependent apoptosis [23]. Rutin, a glycoside found in green tea and apple trees, exhibited synergistic drug-herb interactions in combination with 5-FU, to induce apoptosis in prostate cancer cells [24]. In addition, verbascoside, a phenylethanoid glycoside found in plantago seeds, was found to downregulate the phosphatidylinositol 3-kinases (PI3K)/protein kinase B (Akt) pathway, leading to the sensitization of HCT116 and Caco-2 adenocarcinoma cells to 5-FU [25].

Therefore, in this study, we aimed to evaluate the effects of *C. gigantea* stem bark extracts on the inhibition of growth and induction of apoptosis in colon cancer cells. We also evaluated the effects of combinations of *C. gigantea* stem bark extracts and 5-FU to establish a novel anticancer regimen for future application in cancer therapeutic studies. Moreover, a combination therapy of *C. gigantea* with the minimum dose of 5-FU was hypothesized to improve the suppression of cancer cell proliferation, compared to the monotherapy. The results of this study may provide a basis for the development of new anticancer strategies involving *C. gigantea* combination therapy, which may accelerate the treatment outcomes in cancer. There are only a few reports on the phytochemicals present in *C. gigantea* stem bark and on their anticancer activities. These findings will shed light on the dichloromethane (DCM) fraction of the *C. gigantea* stem bark extract as a valuable source for purification of new potent anticancer agents in the future.

## Materials and methods

### Plant material

Fresh stem barks of *C. gigantea* (L.) Dry and. were collected from the Thoen District, Lampang Province, Thailand, between June, 2015 and April, 2018 (latitude/longitude: 17˚36′9″N/99˚12′50″E). Plant collection in the Lampang Province and use of the collected plants for research purposes was approved according to Plant Varieties Protect Act B.E. 2542 (1999) section 53 under permission number 0278 from the Department of Agriculture, Ministry of Agricultural and Cooperatives, Thailand. The appearance of the flowers, leaves, fruits, and dry stem barks are shown in S1A–S1D Fig, respectively. The herbarium specimen of *C. gigantea* (voucher specimen No. 005194; shown in S1E Fig) was identified by Dr. Pranee Nangngam, a taxonomist and a coauthor, and stored at the PNU Herbarium, Department of Biology, Faculty of Science, Naresuan University, Phitsanulok, Thailand. The fresh stem barks (27 kg) were shade-dried at an ambient temperature (35 ± 7˚C) to obtain 6 kg of dried barks (22.2% yield). The dried stem barks were powdered using a blender and stored in an air-tight plastic bag at room temperature (30 ± 5˚C) until extraction.

### Sample preparation

The dry powdered stem barks of *C. gigantea* (6 kg) were soaked in and extracted with 95% ethanol (20 L × 3 times, AR grade, Thailand) using an ultrasonicator for 1 h at room temperature. The supernatants were combined and the solvent was removed using a rotary evaporator (Buchi, Switzerland) at 45˚C to obtain the 336 g of crude *C. gigantea* ethanolic extract (CGEtOH, dark greenish brown sticky extract, 5.6% yield from dry powder). The CGEtOH extract (100 g) was subjected to liquid–liquid partition between purified water (200 mL) and DCM (300 mL × 3 times, AR grade, LabScan, Thailand). The DCM layers from the three extractions were combined and the solvent was removed using a rotary evaporator at 50˚C to yield the DCM fraction (CGDCM, greenish brown sticky extract, 23.0 g, 23.0% yield from CGEtOH). The water layer was further partitioned with ethyl acetate (EtOAc; 300 mL × 3 times, AR grade, LabScan, Thailand). The EtOAc layers were combined, and the solvent was removed under reduced pressure at 50˚C, to yield the EtOAc fraction (CGEtOAc, dark brown sticky extract, 0.5 g, 0.5% yield from CGEtOH). The remaining water layer was lyophilized to obtain the water fraction (CGW, light brown yellowish powder, 55.2 g, 55.2% yield from CGEtOH). The fractions (CGEtOH, CGDCM, CGEtOAc, and CGW) were stored in a refrigerator (4 ± 3˚C).

## Quantitative analysis of secondary metabolites in *C. gigantea* stem bark extracts

**Total cardiac glycoside content.** The total cardiac glycoside content was determined using a modification of the method described by Tofighi et al. [26]. The sample solution (1 mL, 1 mg/mL in ethanol: water 1:1) was mixed with 1 mL of freshly prepared Baljet's reagent (95 mL of 1% picric acid mixed with 5 mL of 10% NaOH solution). The reaction mixture was left in the dark at room temperature (30 ± 5˚C) for 60 minutes. It was then diluted with purified water (2 mL), and the absorbance was measured at 495 nm using a UV/Vis spectrophotometer (Shimadzu UV-1800, Japan). The total cardiac glycoside content was calculated using a digoxin (Sigma-Aldrich, USA) calibration curve in the range of 5–50 µg/mL ($Y = 0.0154X + 0.05$, $R^2 = 0.9989$, where Y represents the absorbance of digoxin at 495 nm, X represents the concentration of digoxin (µg/mL), and $R^2$ represents the linear correlation coefficient). The cardiac glycoside content was expressed as milligram digoxin equivalents per gram of extract (mg DXE/g extract). The experiments were performed in triplicate.

**Total triterpenoid content.** The total triterpenoid content was analyzed as described by Chang et al. [27] with some modifications. Briefly, 200 µL of the sample solution (1 mg/mL in glacial acetic acid) was mixed with a 5% vanillin-acetic acid solution (1 mL) and sulfuric acid (1.8 mL). The mixture was then heated to 70˚C for 30 minutes, before cooling down to room temperature. Finally, glacial acetic acid (2 mL) was added and mixed well. The absorbance of the reaction mixture was determined at 575 nm. The total triterpenoid content was calculated using an ursolic acid (Tokyo Chemical, Japan) calibration curve in the range of 2–40 µg/mL ($Y = 0.045X - 0.0947$, $R^2 = 0.9996$, where Y represents the absorbance of ursolic acid at 575 nm, X represents the concentration of ursolic acid (µg/mL), and $R^2$ represents the linear correlation coefficient). The results were expressed as milligram ursolic acid equivalents per gram of extract (mg UAE/g extract).

**Total phenolic content.** The total phenolic content in the *C. gigantea* stem bark extracts was analyzed using a modification of the method described by Baba and Malik [28]. The sample solution (1 mL, 1 mg/mL) was mixed with 1 mL of 1:10 Folin-Ciocalteu reagent (Merck, Germany) and vortexed for 5 minutes. Then, 1 mL of saturated sodium bicarbonate (60 g/L) was added, and the mixture was allowed to stand for 90 min in the dark at ambient temperature for color development. The absorbance of the mixture was then determined at 725 nm using a UV/Vis spectrophotometer (Shimadzu UV-1800, Japan). Gallic acid (1.7–13.3 µg/mL, Sigma-Aldrich, China) was used as the standard ($Y = 0.1441X - 0.0682$, $R^2 = 0.9974$, where Y represents the absorbance of gallic acid at 725 nm, X represents the concentration of gallic acid (µg/mL), and $R^2$ represents the linear correlation coefficient). The total phenolic content of the extracts was expressed as milligram gallic acid equivalents per gram of extract (mg GAE/g extract).

**Total flavonoid content.** The total flavonoid content was analyzed using a modification of the method described by da Silva et al. [29]. In brief, 1 mL of the test sample solution (1 mg/mL in 50% ethanol in water) was mixed with 1 mL of 2% aluminum chloride in methanol. The mixture was left in the dark at room temperature for 25 minutes before the absorbance was measured at 415 nm. The total flavonoid content (n = 3) was calculated using a rutin (Sigma-Aldrich, USA) calibration curve in the range of 10–100 µg/mL ($Y = 0.0253X + 0.0159$, $R^2 = 0.9980$, where Y represents the absorbance of rutin at 415 nm, X represents the concentration of rutin (µg/mL), and $R^2$ represents the linear correlation coefficient). The results were expressed as milligram rutin equivalents per gram of extract (mg RTE/g extract).

**Total alkaloid content.** The total alkaloid content was analyzed using a modification of the method described by Patel et al. [30]. The sample (10 mg) was dissolved in 1 mL of 2 N

HCl, and extracted into chloroform 3 times (3, 3, and 4 mL). The chloroform layers were discarded. The HCl layers were neutralized with 0.1 N NaOH, before adding 5 mL bromocresol green solution and 5 mL phosphate buffer solution (pH 4.7). This mixture was extracted thrice with chloroform (3, 3, and 4 mL). The chloroform layers were combined and the absorbance was measured at 420 nm. The absorbance was used to calculate the total alkaloid content by comparing with a berberine chloride (Sigma Aldrich, USA) standard curve, in the range of 2–16 µg/mL (Y = 0.0342X – 0.0223, $R^2$ = 0.9946, where Y represents the absorbance of berberine chloride at 420 nm, X represents the concentration of berberine chloride (µg/mL), and $R^2$ represents the linear correlation coefficient). The total alkaloid content was expressed as milligram berberine chloride equivalents per 10 g extract (mg BCE/10 g extract).

## HPLC analysis of the *C. gigantea* stem bark extracts

The *C. gigantea* stem bark extracts (CGEtOH, CGDCM, CGEtOAc, and CGW) were analyzed by high-performance liquid chromatography (HPLC). Calotropin, one of the major cardenolides present in *C. gigantea*, was used as a standard. This was obtained as a gift from Professor Zhi-Hong Jiang and Dr. Li-Ping Bai, Macau University of Science and Technology, Macau. The standard was characterized elsewhere [4] and its identity was reconfirmed by high resolution mass spectroscopy ($C_{29}H_{40}O_9$, exact mass = 532.2672, *m/z* of [M+HCOO]$^-$ (found in the analysis) = 577.2651; difference 3.7 ppm) using a mass spectrometer (an Agilent 6540 UHD Accurate-Mass quadrupole time-of-flight liquid chromatograph mass spectroscopy, Agilent Technologies) equipped with dual electrospray ionization in the negative mode (*m/z* range 200–800). The nebulizer pressure ($N_2$) was set at 30 psi, the drying gas flow rate was 10 L/min, and the drying gas temperature was 350˚C. The HPLC analysis of the *C. gigantea* bark extracts was carried out using a modification of the method described by Kharat and Kharat [31]. In brief, the test samples (20 µL, 5 mg/mL in methanol) were injected into the HPLC system (Shimadzu pump LC-10ATvp, Japan) and the mobile phase was passed at a flow rate of 1 mL/min for 15 minutes; detection was performed using an ultraviolet/visible detector (222 nm). A Phenomenex Luna® C18(2) (150 mm × 4.6 mm, 3 µm) with a guard column (Phenomenex C18, 4 mm × 3 mm, 5 µm) was used as the stationary phase. The isocratic mobile phase consisted of 55% methanol (HPLC grade, LabScan, Thailand) in water. The retention time of calotropin was approximately 7.43 ± 0.08 min. The calotropin content in each extract was calculated from the calotropin standard curve (0.2–100 µg/mL, Y = 34377X - 43075, $R^2$ = 0.9991, where Y = peak area at a retention time of 7.43 ± 0.08 min and X = concentration of calotropin (µg/mL)). The results were expressed as milligram calotropin per gram of extract (mean ± standard deviation [SD] of three independent experiments).

## Cell culture

Human colorectal carcinoma HCT116 (CCL-247, ATCC, USA) and colorectal adenocarcinoma HT-29 (HTB-38, ATCC, USA) cells were cultured in McCoy's medium (Corning, USA), supplemented with 10% fetal bovine serum (FBS; Gibco, USA) and 1% Antibiotic-Antimycotic solution (Gibco, USA). Cells were incubated at 37˚C in a humidified atmosphere with 5% $CO_2$. The culture medium was replaced every 2 days. Human foreskin fibroblast, HFF-1 (SCRC-1041, ATCC, USA), cells were cultured in Dulbecco's modified Eagle's medium (Gibco, USA), supplemented with 15% FBS (Gibco, USA). The cells were maintained in a humidified incubator at 37˚C under a humidified atmosphere with 5% $CO_2$. The medium was replaced every 2 days. When cell confluence reached 80%–90%, the cells were subcultured.

## Evaluation of cell viability by MTT assay

Cells were seeded at a density of 20,000 cells/well in 96-well plates (SPL, Korea), and incubated at 37°C for 24 h. They were treated with the four fractions of the extract, each dissolved in 0.8% DMSO. 5-FU (Sigma, Japan) was used as the positive control. After treatment, the cells were incubated with 2 mg/mL 3-(4,5-dimethylthiazol-2-yl)-2,5-diphenyltetrazolium bromide (MTT) solution (Merck, Germany) at 37°C for 4 h. The mitochondrial reductase enzyme converts MTT (yellow color) into formazan crystals (purple). DMSO was added to dissolve these crystals and the optical density (OD) of the samples was measured at 595 nm using a microplate reader (Biotek, USA). The fraction exhibiting the highest cytotoxic potency was used in further tests to explore synergistic effects with 5-FU.

## Evaluation of cell migration by wound healing assay

Cells were seeded at a density of $2.5 \times 10^5$ cells/well in a 12-well plate to attain a monolayer with up to 60% confluence within 48 h. Then, wounds (cell-free scratched area with linear-edged lines and a consistent gap width) were made at the center of each well. The cells were then treated with CGDCM, 5-FU, or a combination of both. Cell migration from the wound edges to the wound gap was visualized with an inverted optical microscope (IX71, Olympus, Japan), and quantified by measuring the wound distance using the cellSens Standard [Ver.2.3] software. Images were taken at ×10 magnification.

## Detection of apoptotic cells by nuclear staining

Cells were treated with CGDCM, 5-FU, or a combination of the two for 24 h. The cells were then harvested and fixed with 10% formalin for 15 min at room temperature. After fixation onto a glass slide (Thermo Fisher Scientific, USA), the cells were stained with 4′,6-diamidino-2-phenylindole dihydrochloride (Invitrogen, USA) at room temperature for nuclear staining, and visualized by fluorescence microscopy (BX53F2, Olympus Corporation, Japan).

## Evaluation of mitochondrial membrane potential (ΔΨm)

Cells were treated with CGDCM, 5-FU, or a combination of both for 24 h. The cells were then harvested and fixed with 10% formalin for 15 min at room temperature. After fixation onto a glass slide, the cells were stained with 5,5′,6,6′-tetrachloro-1,1′,3,3′-tetraethyl-imidacarbocyanine iodide, 5,5′,6,6′-tetrachloro-1,1′,3,3′-tetraethylbenzimidazolocarbo-cyanine iodide (JC-1) dye (Invitrogen, USA), a probe for mitochondrial membrane potential, at room temperature. Red fluorescence from the cells was indicative of normal ΔΨm, with JC-1 aggregation in mitochondria. Depolarization of ΔΨm was expressed as green fluorescence, reflecting the presence of monomeric JC-1 in the cytosol. The stained cells were analyzed by fluorescence microscopy.

## Evaluation of apoptosis by flow cytometry

After being treated with CGDCM, 5-FU, or a combination of both for 24 h, cells were harvested and stained with Annexin V and Dead Cell Assay Kit (MCH100105, Merck, Germany), as per the manufacturer's instructions. Stages of apoptotic cell death were determined by double staining with annexin V and dead cell marker, 7-amino-actinomycin D (7-AAD). Annexin-V stains phosphatidylserine in the early stages of apoptosis, whereas both annexin V and 7-AAD stain cells in the late apoptotic stages. Briefly, cells in 1% FBS phosphate-buffered saline were stained with annexin V and 7-AAD and incubated for 20 min in the dark. Apoptotic cells were analyzed using Muse Cell Analyzer (0500–3115, Merck, Germany).

## ROS quantification

Cells were treated with CGDCM, 5-FU, or a combination of both for 24 h. CM-H2DCFDA, a chloromethyl derivative of H2DCFDA (2′,7′-dichlorodihydrofluorescein diacetate) (Thermo Fisher Scientific, USA) was then added to each sample and the mixtures were incubated at 37˚C in the dark under an atmosphere of 5% $CO_2$. Cell staining indicated the accumulation of intracellular ROS. The cells were visualized using a fluorescence microscope.

## Intracellular adenosine triphosphate (ATP) assay

Following treatment with CGDCM, 5-FU, or a combination of both for 24 h, the intracellular ATP levels were determined using the ATP assay kit (Elabscience, USA), as per the manufacturer's instructions. The cellular ATP content was evaluated by measuring the OD at 636 nm using a microplate reader.

## Western blot analysis

Cells seeded in a 35 mm culture dish were cultured for 24 h and allowed to reach 60% confluence. The cells were harvested, and the intracellular proteins were extracted using Mammalian Protein Extraction Reagent (M-PER; Thermo Fisher Scientific, USA), containing a proteinase inhibitor cocktail (HIMEDIA, India). Proteins from the cell lysate were collected and quantified by addition of bicinchoninic acid assay reagent (Thermo Fisher Scientific, USA), and the OD was measured at 590 nm using a microplate reader. Equal amounts of proteins were separated by sodium dodecyl sulfate-polyacrylamide gel electrophoresis and transferred onto a polyvinylidene fluoride membrane. After incubation with a blocking solution (GeneDireX, USA), the membranes were incubated with anti–B-cell lymphoma 2 (Bcl-2) (Thermo Fisher Scientific, USA) and anti-cleaved caspase-3 (Cell Signaling Technology, USA) primary antibodies, and then exposed to horseradish peroxidase-conjugated goat anti-rabbit or anti-mouse secondary antibodies (Life Technologies, Invitrogen). β-actin (Cell signaling Technology, USA) was used as an internal standard. Protein bands were visualized using Luminata Forte Western HRP Substrate (Merck Millipore, USA) and detected by Chemiluminescence western blot detection (Image Quant LAS 4000; GE Healthcare Life Sciences, USA). Relative expression levels (%) of proteins vis-à-vis β-actin expression were calculated using the Image J software version 1.46.

## Statistical analysis

Data from three independent experiments are shown as mean ± SD. One-way analysis of variance (ANOVA) or Student's *t*-test with Tukey's post hoc analysis was used to determine the statistical significance of differences between the experimental and control groups. Differences were considered statistically significant at $p < 0.05$. Data analysis was done using the Graph Prism Software version 9.

# Results

## Preparation of *C. gigantea* stem bark extracts

Dry *C. gigantea* stem barks were obtained with a yield of 22.2% after shade-drying fresh barks. Ultrasound-assisted extraction was done in 95% ethanol to obtain the total ethanolic extract (CGEtOH) with a yield of 5.6% from the dry bark. The CGEtOH was subsequently fractionated by liquid–liquid extraction to obtain CGDCM, CGEtOAc, and CGW fractions, with yields of approximately 23.0%, 0.5%, and 55.2%, respectively. Fractionation separated the phytochemicals in the crude extract into three fractions, based on their polarity. CGDCM

contained the highest amount of non-polar compounds, followed by CGEtOAc, and CGW with the largest amount of polar compounds. The yield of CGW was the highest, followed by that of CGDCM, and CGEtOAc, which had a very low yield. The crude extract and the three fractions were evaluated for their phytochemical content and apoptotic activity in colon cancer cells.

## Quantification of secondary metabolites and calotropin in the *C. gigantea* stem bark extracts

**Total cardiac glycoside content.** *C. gigantea* is thought to be rich in cardiac glycosides. A number of cardenolides, such as uscharin, calactin, calotropin, and calotoxin, have been isolated from this plant and shown to exhibit anticancer activity [4,6,11]. The total cardiac glycoside content in each of the extracts is summarized in Table 1. CGEtOAc was found to have the highest total cardiac glycoside content (176.1 mg DXE/g extract), whereas CGW was found to have the lowest (35.7 mg DXE/g extract). The total cardiac glycoside content of CGDCM was similar to that of CGEtOH (91.7 and 77.2 mg DXE/g extract, respectively).

**Total triterpenoid content.** The total triterpenoid content of the four extracts is presented in Table 1. Ursolic acid was used as the standard. CGDMC was found to have a significantly higher triterpenoid content (632.3 mg UAE/g extract) than the other fractions, probably owing to their non-polar nature [9,32]. The total triterpenoid content of CGEtOAc was higher than that of CGEtOH (95.5 and 58.9 mg UAE/g extract, respectively), whereas CGW had the lowest content (29.2 mg UAE/g extract).

**Total phenolic content.** The *C. gigantea* stem bark extracts contained various phenolic compounds. CGEtOAc had a higher phenolic content than CGDCM and CGEtOH. Phenolic contents were in the range of 16.2–29.8 mg GAE/g extract, consistent with previous reports on the phenolic content of *C. gigantea* flowers [33]. Surprisingly, the most polar fraction, CGW, had the lowest phenolic content (4.6 mg GAE/g extract). The phenolic content of all the tested extracts is illustrated in Table 1.

**Total flavonoid content.** The total flavonoid content of all the extracts is summarized in Table 1. Among the four samples, CGDCM was found to have the highest flavonoid content, followed by CGEtOH, CGEtOAc, and CGW (47.9, 36.7, 22.9, and 17.1 mg RTE/g extract, respectively). The results were in agreement with those of previous studies investigating the flavonoid content of extracts of the aerial parts of *C. gigantea* [28,33].

**Total alkaloid content.** Alkaloids have been reported in several parts of *C. gigantea* [34]. The results obtained in the present study are depicted in Table 1. CGW was found to have the highest alkaloid content (112.0 mg BCE/10 g extract), whereas CGEtOH, CGDCM, and CGEtOAc had similar content, ranging between 35.0 and 47.8 mg BCE/10 g extract.

**Calotropin content.** Calotropin is a promising anticancer cardenolide found in *C. gigantea*. It was chosen as a standard for the standardization of the four extracts. Calotropin was

**Table 1. Content of secondary metabolites and calotropin in the *Calotropis gigantea* stem bark extracts.**

| Extracts | Cardiac glycosides (mg DXE/g extract) | Triterpenoids (mg UAE/g extract) | Phenolics (mg GAE/g extract) | Flavonoids (mg RTE/g extract) | Alkaloids (mg BCE/10 g extract) | Calotropins (mg/10g extract) |
|---|---|---|---|---|---|---|
| **CGEtOH** | 77.2 ± 5.84 | 58.9 ± 15.17 | 16.2 ± 0.08 | 36.7 ± 1.20 | 47.8 ± 1.98 | 9.5 ± 1.27 |
| **CGDCM** | 91.7 ± 6.21 | 632.6 ± 31.29 | 19.7 ± 0.61 | 47.9 ± 0.40 | 35.7 ± 1.39 | 6.2 ± 0.32 |
| **CGEtOAc** | 176.1 ± 0.94 | 95.5 ± 5.25 | 29.8 ± 0.66 | 22.9 ± 0.39 | 35.0 ± 0.84 | 2.7 ± 0.06 |
| **CGW** | 35.7 ± 2.18 | 29.2 ± 4.01 | 4.6 ± 0.25 | 17.1 ± 0.44 | 112.0 ± 1.91 | 8.6 ± 0.15 |

Abbreviations: CGEtOH, *C. gigantea* ethanolic extract; CGDCM, *C. gigantea* dichloromethane extract; CGEtOAc, *C. gigantea* ethyl acetate extract; CGW, *C. gigantea* water extract; DXE, digoxin equivalent; UAE, ursolic acid equivalent; GAE, gallic acid equivalent; RTE, rutin equivalent; BCE, berberine chloride equivalent.

isolated and purified from the latex of *C. gigantea*, and characterized, as described in a previous report [4]. The HPLC chromatogram of calotropin is illustrated in S2A Fig, and shows a peak at a retention time of approximately 7.43 ± 0.08 min. Its accurate mass was confirmed by high resolution mass spectroscopy, as shown in S2B Fig ($m/z$ actual $[M + HCOO]^-$ = 577.2651 vs. exact mass = 532.2672; difference of approximately 3.7 ppm). This suggested the molecular formula of the compound to be $C_{29}H_{40}O_9$. The HPLC chromatograms of CGEtOH, CGDCM, CGEtOAc, and CGW, analyzed according to the method described by Kharat and Kharat [31], are shown in S2C–S2F Fig. Various peaks were observed. The area of the peak at retention time 7.3–7.5 min was used to calculate the calotropin content of each sample. The calotropin content of the extracts is summarized in Table 1 (n = 3) and S2F Fig. CGEtOH was found to have the highest calotropin content, followed by CGW, CGDCM, and CGEtOAc (9.5, 8.6, 6.2, and 2.7 mg calotropin/10 g extract, respectively).

**Cytotoxic effects of *C. gigantea* stem bark extracts on HCT116 and HT-29 cells.** The cytotoxic effects of the four extracts (CGEtOH, CGDCM, CGEtOAc, and CGW) were evaluated in an MTT assay using HCT116 and HT-29 cells. Cells were treated with different concentrations of the extracts for 24 h. The viability of the HCT116 and HT-29 cells decreased in a dose-dependent manner after treatment with each of the extracts. The half maximal inhibitory concentration ($IC_{50}$) was 5.9, 7.7, 32.8, and 39.0 μg/mL for CGDCM, CGEtOAc, CGEtOH, and CGW, respectively, in HCT116 cells. In HT-29 cells, the $IC_{50}$ values were 44.0, 43.6, 44.0, 60.6, and 86.7 μg/mL for CGDCM, CGEtOAc, CGEtOH, and CGW, respectively (Fig 1A and 1B). CGDCM and CGEtOAc exhibited the highest potency in both the cell lines. 5-FU exhibited dose-dependent cytotoxicity, with an $IC_{50}$ of 248.1 μM (34.1 μg/mL) in HCT116 cells, and 3598 μM (457 μg/mL) in HT-29 cells (Fig 1C and1D). A summary of the $IC_{50}$ values of the fractions from *C. gigantea* stem bark extracts on HCT116 and HT-29 cells at 24 h is demonstrated in Table 2. These findings are consistent with those of previous studies, wherein it was found that 5-FU produced 60% inhibition in HCT116 cells treated at 100 μM for 24 h [35]. Incubation with 50 μM 5-FU for 24 h induced ~20% growth inhibition in HCT116 cells [36]. The $IC_{50}$ value of 5-FU for 48 h treatment of HCT116 cell was 10 μg/mL [37].

On the basis of these findings, sub-$IC_{50}$ and supra-$IC_{50}$ concentrations of CGDCM and CGEtOAc (1, 2, 4, 8, and 10 μg/mL) were selected for use in combination with 5-FU (5 μM or 0.65 μg/mL) in further experiments. These findings were supported by those of a previous study, wherein a low concentration (20 μM) of 5-FU had weak cytotoxicity in HCT116 cells after 24 h of incubation [38]. This concentration of 5-FU was selected because it produced the minimum significant response in HCT116 cells.

**The combination effects of CGDCM and CGEtOAc with 5-FU on the cytotoxicity in HCT116 cells.** Following a 24 h incubation period, it was found that combinations of CGDCM (1 or 2 μg/mL) with 5-FU (5 μM or 0.65 μg/mL) did not exhibit cytotoxicity in HCT116 cells (Fig 2A). Combinations of 4, 8, and 10 μg/mL CGDCM with 5-FU (5 μM or 0.65 μg/mL) significantly decreased the cell viability, compared with 5-FU alone. Combinations of CGDCM at 8 and 10 μg/mL with 5-FU (5 μM or 0.65 μg/mL) exhibited combination effects, showing greater inhibition of cell viability, compared with either of the treatments alone. However, the cytotoxicity of CGDCM (8 μg/mL) in combination with 5-FU was not significantly different from that of a combination of CGDCM (10 μg/mL) and 5-FU. Cells were incubated for 48 h with CGDCM, 5-FU (5 μM or 0.65 μg/mL), or a combination of both; potent cytotoxic effects were observed, but no combination effects were observed (Fig 2B). As seen in Fig 2C, treatment with a combination of CGEtOAc (1, 2, 4, 8, and 10 μg/mL) and 5-FU (5 μM or 0.65 μg/mL) for 24 h showed a lower potency in reducing cell viability than did CGDCM. After incubation for 48 h, treatment with CGEtOAc, 5-FU, and a combination of both exhibited a similar efficacy to that of CGDCM at 48 h (Fig 2D). Therefore, combinations

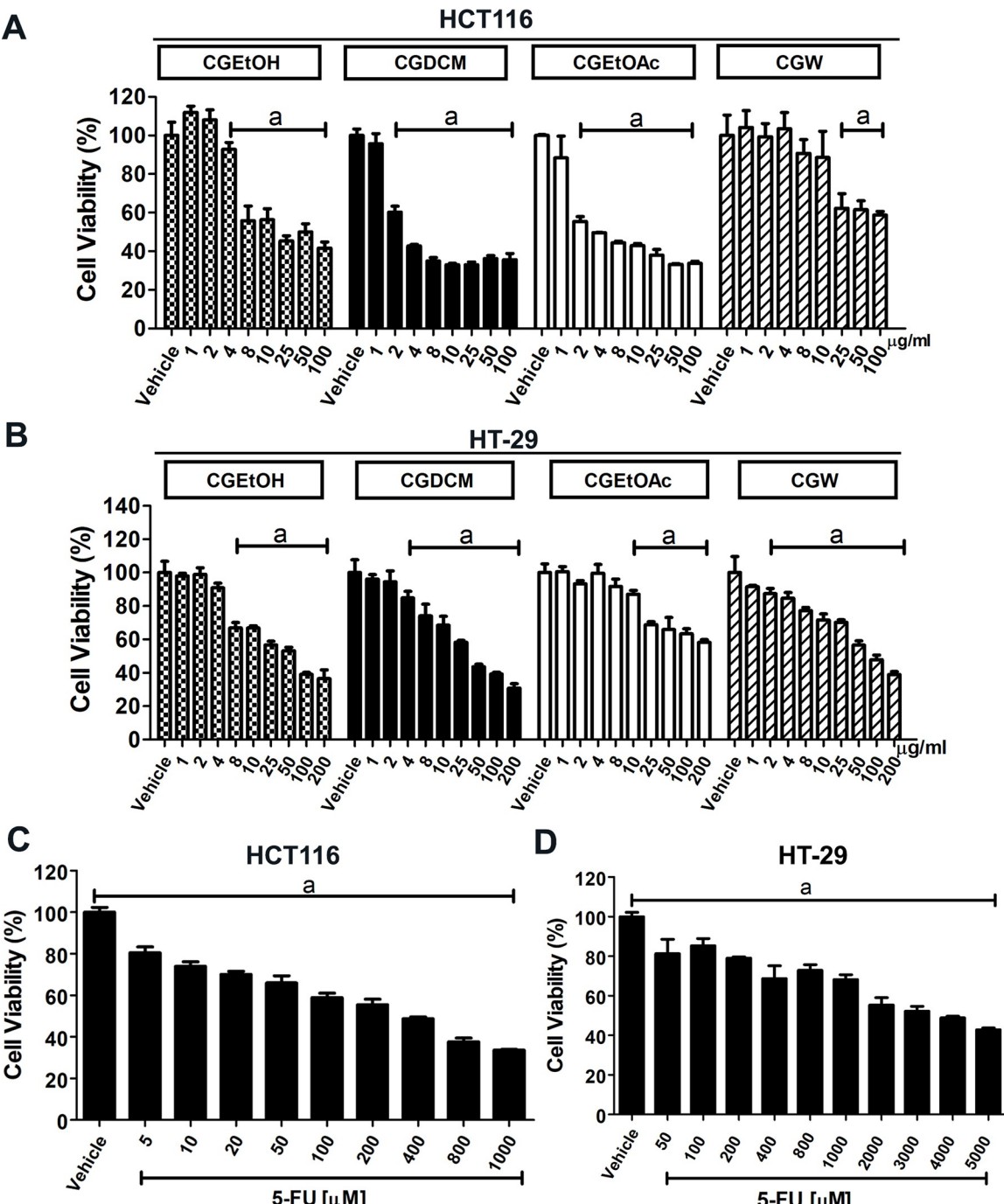

**Fig 1. Cytotoxic effects of *Calotropis gigantea* stem bark extracts on HCT116 and HT-29 cells.** The viability of HCT116 (**A**) and HT-29 (**B**) cells, evaluated by MTT assay. The vehicle control group was 0.8%DMSO. The effect of the positive control, 5-FU, on HCT116 (**C**) and HT-29 (**D**) cells. The data are presented as mean ± SD from a minimum of three independent experiments, and were analyzed using one-way ANOVA with Tukey's HSD test. *p < 0.05 compared with the vehicle. Abbreviations: CGEtOH, *C. gigantea* ethanolic extract; CGDCM, *C. gigantea* dichloromethane extract; CGEtOAc, *C. gigantea* ethyl acetate extract; CGW, *C. gigantea* water extract; 5-FU, 5-fluorouracil.

**Table 2. IC$_{50}$ values of fractions from *Calotropis gigantea* stem bark extracts on HCT116 and HT-29 cells at 24 h of treatment.**

| Extracts | IC50 (μg/mL) | |
|---|---|---|
| | **HCT116** | **HT-29** |
| **CGEtOH** | 32.8 ± 0.83 | 60.6 ± 3.61 |
| **CGDCM** | 5.9 ± 0.62 | 44.0 ± 4.06 |
| **CGEtOAc** | 7.7 ± 0.89 | 43.6 ± 9.34 |
| **CGW** | 39.0 ± 8.88 | 86.7 ± 10.81 |
| **5-FU** | 34.1 ± 3.57 (248.1 ± 29.62 μM) | 457.1 ± 29.63 (3,598.0 ± 199.83 μM) |

Abbreviations: CGEtOH, *C. gigantea* ethanolic extract; CGDCM, *C. gigantea* dichloromethane extract; CGEtOAc, *C. gigantea* ethyl acetate extract; CGW, *C. gigantea* water extract; 5-fluorouracil, 5-FU.

of CGDCM (4, 8, and 10 μg/mL) with 5-FU (5 μM or 0.65 μg/mL) and an incubation period of 24 h were selected for apoptosis induction experiments in HCT116 cells.

Results of the wound healing assay supported the findings for induction of cell death by CGDCM and a combination of CGDCM with 5-FU in HCT116 cells (Fig 2E and 2F). Suppression of cell migration to fill the wound gap is a potential therapeutic approach for the treatment of cancer [39]. The gap distance remained unchanged after treatment for 12, 24, and 48 h with CGDCM (4, 8, 10 μg/mL) in combination with 5-FU (5 μM or 0.65 μg/mL), suggesting inhibition of cell proliferation and migration in the HCT116 cell line. Moreover, normal HFF-1 cells were found to be resistant to CGDCM (4, 8, 10 μg/mL), 5-FU, and their combinations, upon treatment for 24 h (Fig 2G). This is consistent with the findings in a previous study that 5-FU (5 μM or 0.65 μg/mL) exhibited slight cytotoxicity in normal cells [35]. The results suggest that CGDCM may be selectively cytotoxic toward cancer cells.

**Effect of CGDCM and 5-FU combination therapy on apoptosis in HCT116 cells.** The cytotoxic effect of CGDCM, 5-FU, and a combination of the two in the induction of the apoptosis pathway was analyzed and is shown in Fig 3A and 3B. Treatment with 5-FU (5 μM or 0.65 μg/mL) for 24 h did not significantly increase apoptosis. In contrast, apoptosis was induced in a significantly larger number of cells treated with 4, 8, and 10 μg/mL CGDCM than in those treated with the vehicle. Combinations of CGDCM (4, 8, and 10 μg/mL) with 5-FU (5 μM or 0.65 μg/mL) significantly increased the induction of apoptosis compared with either of the drugs used alone. CGDCM produced the same apoptotic effect at concentrations of 8 and 10 μg/mL, in combination with 5-FU (5 μM or 0.65 μg/mL).

It has been observed in several studies that dissipation of ΔΨm serves as a central controller of the apoptotic cascade in cancer cells, following treatment with anticancer compounds [40]. Analysis of ΔΨm (Fig 4) showed that HCT116 cells treated with CGDCM (4, 8, and 10 μg/mL), 5-FU, and combinations of the two, exhibited higher green and lower red fluorescence than those treated with the vehicle, indicating depolarization of the mitochondrial membrane. These findings suggest that CGDCM, 5-FU, and combination of the two induce apoptosis through a mitochondria-dependent pathway in HCT116 cells.

Next, we evaluated the effects of CGDCM, 5-FU, and their combinations on the expression of Bcl-2 and cleaved caspase 3. The results are summarized in Fig 5A–5D. It was found that the expression of cleaved-caspase 3 increased significantly following combination treatments, compared with the increase upon treatment with CGDCM or 5-FU alone. Conversely, Bcl-2 levels in the cells decreased significantly upon combination treatments, compared with the levels upon treatment with 5-FU alone. Therefore, CGDCM and 5-FU exhibited combination effects in the induction of apoptosis through the upregulation of cleaved-caspase 3 and the downregulation of Bcl-2 expression.

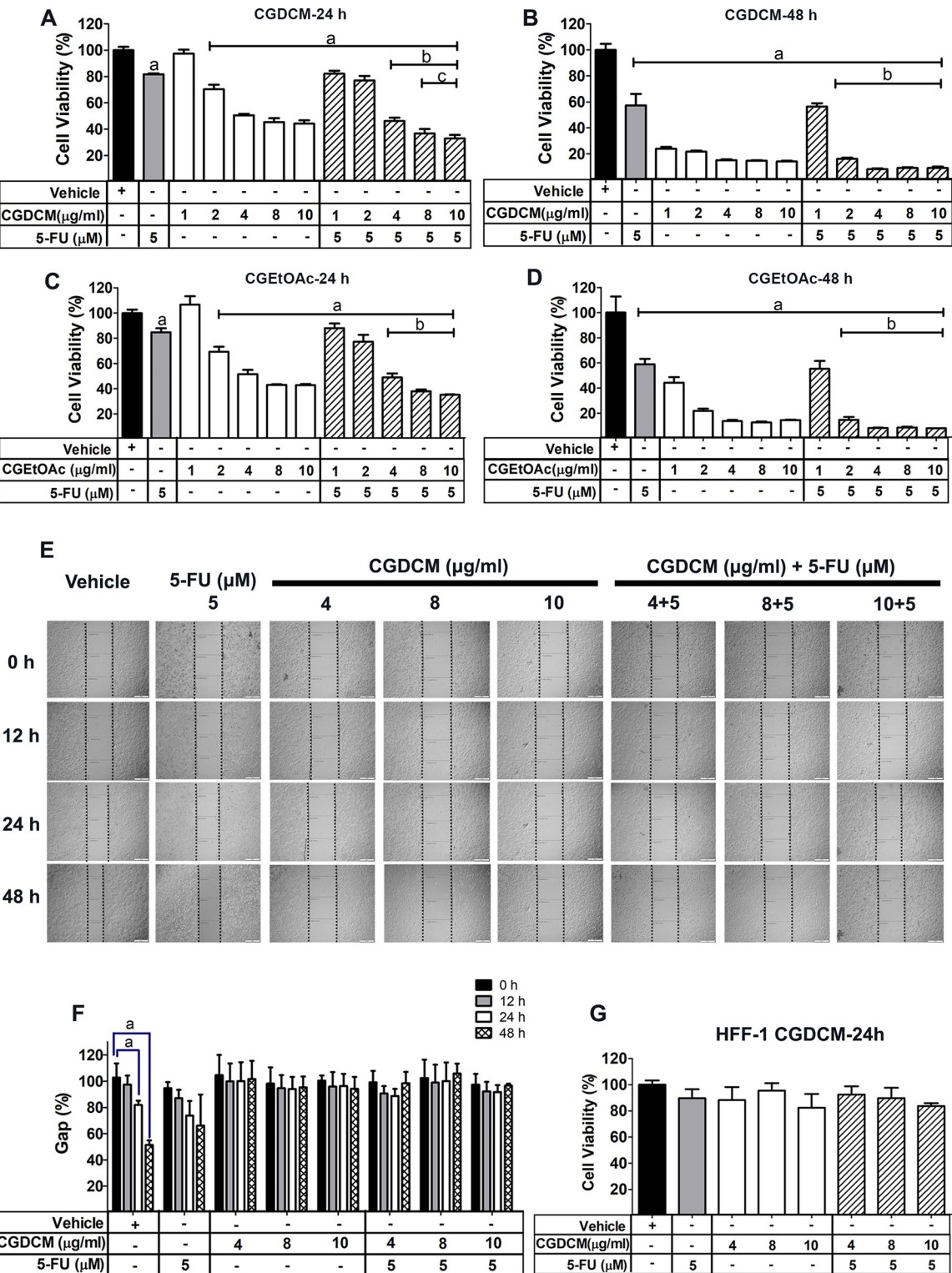

**Fig 2. Combination effects of *Calotropis gigantea* stem bark extracts and 5-FU on cytotoxicity in HCT116 cells.** The viability of HCT116 cells was analyzed in an MTT assay for 24 h (**A**) and 48 h (**B**), following treatment with CGDCM (with/without 5-FU), and for 24 h (**C**) and 48 h (**D**), following treatment with CGEtOAc (with/without 5-FU). Anti-proliferative and anti-migration activities of CGDCM and 5-FU were evaluated in a wound healing assay for 48 h; bars = 500 μm (**E** and **F**). Data are presented as mean ± SD. Cytotoxicity was evaluated in HFF-1 cells for 24 h (**G**). The data were analyzed by one-way ANOVA with Tukey's HSD test. a; $p < 0.05$ vs. vehicle group. b; $p < 0.05$ vs. 5-FU group. c; $p < 0.05$ vs. treatment with CG extract alone. Abbreviations: CGDCM, *C. gigantea* dichloromethane extract; CGEtOAc, *C. gigantea* ethyl acetate extract; 5-FU, 5-fluorouracil.

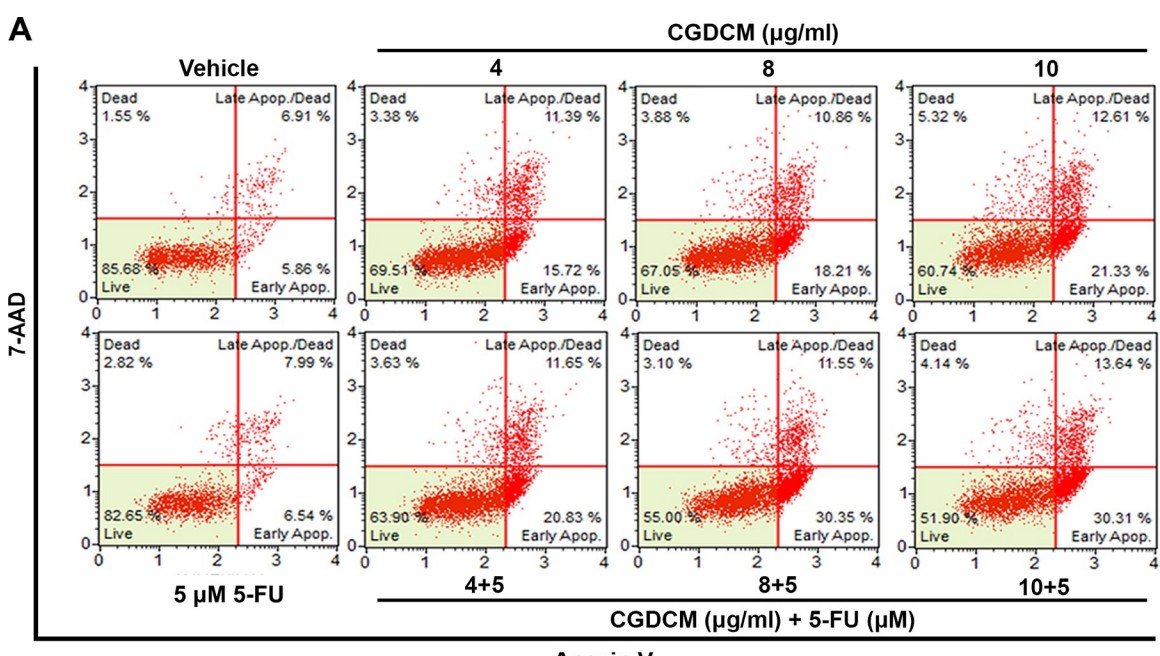

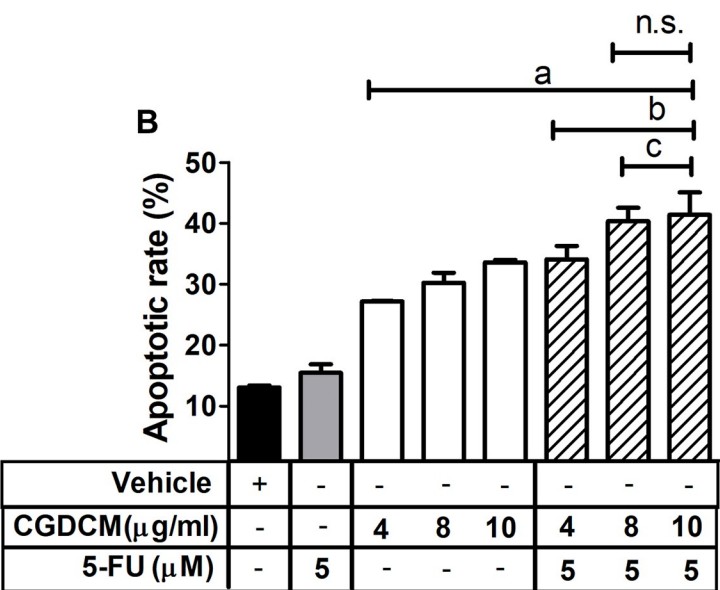

**Fig 3. Combination effects of CGDCM and 5-FU in the induction of apoptosis in HCT116 cells.** The rate of apoptosis in HCT116 cells treated with CGDCM, 5-FU, and combinations of CGDCM (4, 8, and 10 μg/mL) with 5-FU (5 μM or 0.65 μg/mL) for 24 h. The total number of apoptotic (early and late) cells was evaluated by double staining with annexin-V and 7-AAD (**A**) and calculated as a percentage of the total cells (**B**). DMSO (0.8%) was used as the vehicle. The data are presented as mean ± SD, and were analyzed by one-way ANOVA with Tukey's HSD test. a; $p < 0.05$ vs. vehicle group. b; $p < 0.05$ vs. 5-FU group. c; $p < 0.05$ vs. treatment with CGDCM alone. Abbreviations: CGDCM, *C. gigantea* dichloromethane extract; 5-FU, 5-fluorouracil; 7-ADD, 7-amino-actinomycin D.

**Combinations of CGDCM and 5-FU induced apoptosis through the regulation of ATP and ROS production in HCT116 cells.** To understand the mechanism of apoptosis-mediated cell death, it is important to analyze the changes in intracellular ATP production, which is one of the causes of apoptosis in HCT116 cells. ATP plays an important role in the regulation

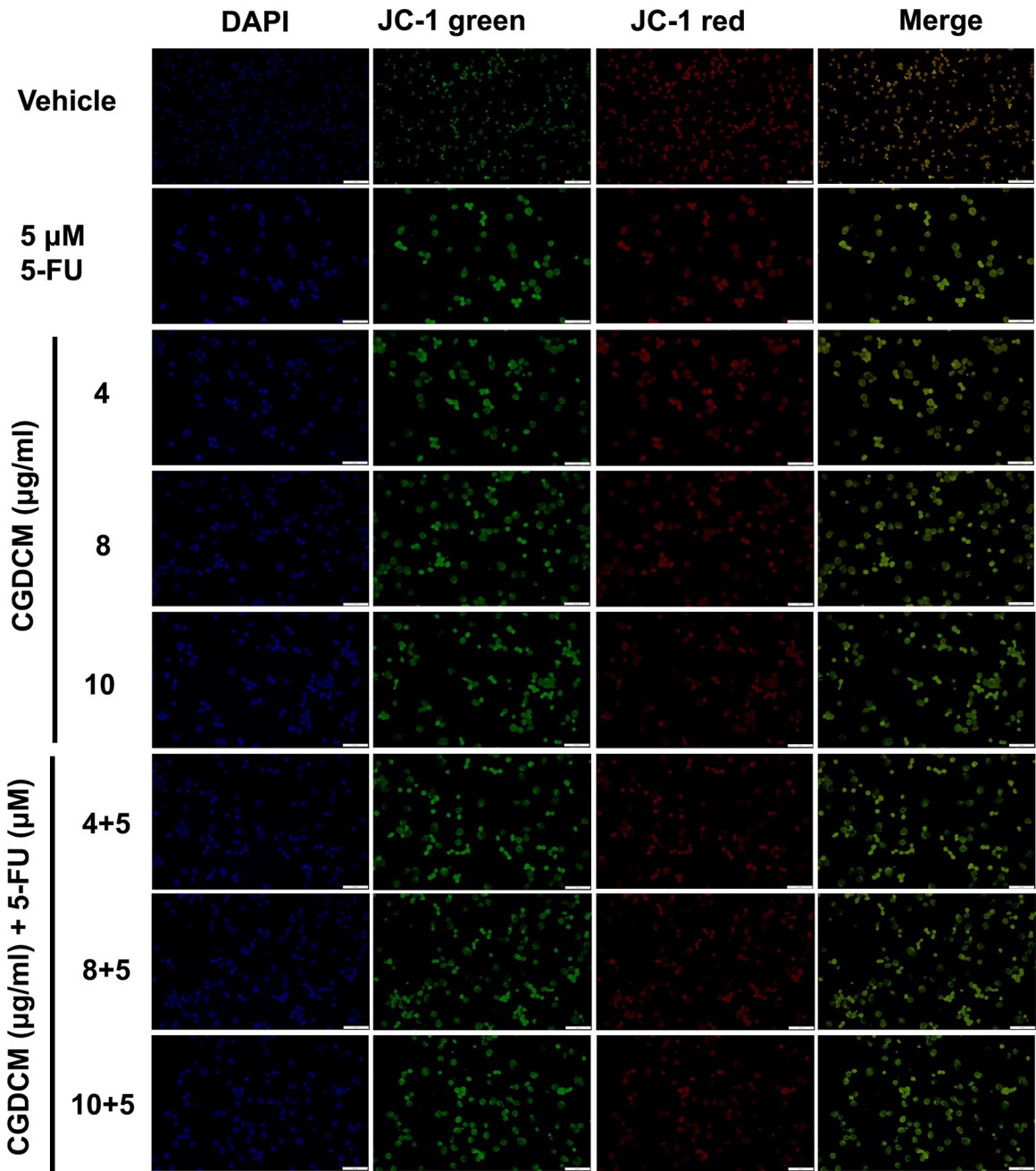

**Fig 4. Combination effects of CGDCM and 5-FU on mitochondrial membrane potential of HCT116 cells.** Cells were treated with CGDCM, 5-FU, and combinations of CGDCM (4, 8, and 10 μg/mL) with 5-FU (5 μM or 0.65 μg/mL) for 24 h. JC-1 staining was visualized by fluorescence microscopy; bars = 100 μm. Abbreviations: CGDCM, *C. gigantea* dichloromethane extract; 5-FU, 5-fluorouracil; DAPI, 4′,6-Diamidino-2-phenylindole dihydrochloride.

of apoptosis in many cancer cells [41,42]. Evaluation of the apoptotic effect revealed that CGDCM (4 and 8 μg/mL) and 5-FU were more potent when used in combination. The drug combination was further analyzed for the mechanisms involved in the induction of apoptosis. CGDCM (8 μg/mL) in combination with 5-FU (5 μM or 0.65 μg/mL) was found to significantly reduce the cellular ATP levels in HCT116 cells after treatment for 24 h (Fig 6A).

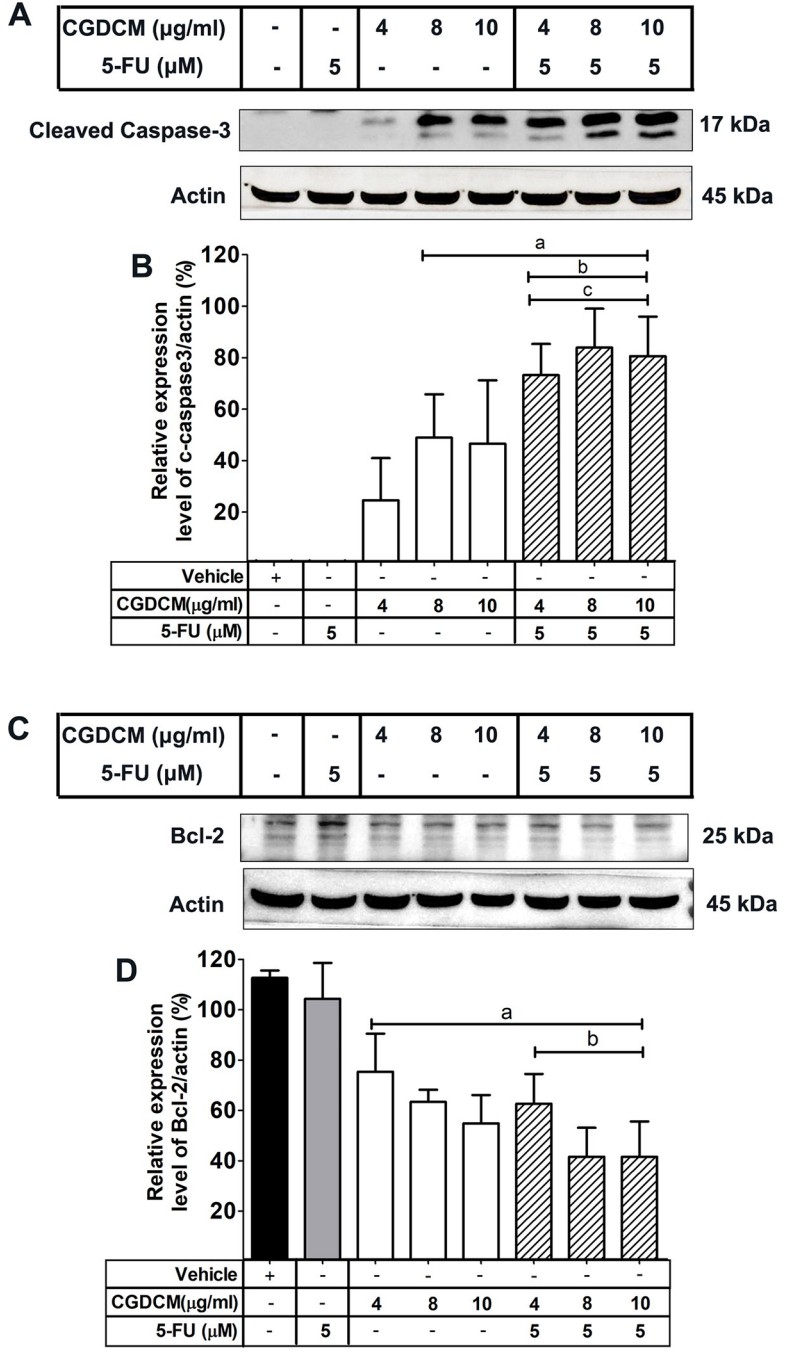

**Fig 5. Effects of CGDCM and 5-FU on the expression of apoptosis proteins in HCT116 cells.** The protein expression levels in HCT116 cells treated with CGDCM, 5-FU, and combinations of CGDCM (4, 8, and 10 µg/mL) with 5-FU (5 µM or 0.65 µg/mL) for 24 h, were analyzed by western blotting (**A–D**). The data are presented as mean ± SD, and were analyzed by one-way ANOVA with Tukey's HSD test. a; $p < 0.05$ vs. vehicle group. b; $p < 0.05$ vs. 5-FU group. c; $p < 0.05$ vs. treatment with CGDCM alone. Abbreviations: CGDCM, *C. gigantea* dichloromethane extract; 5-FU, 5-fluorouracil; Bcl-2, B-cell lymphoma 2.

Increased ROS production has been reported to cause DNA damage, which leads to apoptosis in cancer cells [43]. In the present study, we focused on the correlation between ROS production and the apoptotic effect of CGDCM in combination with 5-FU. The intensity of

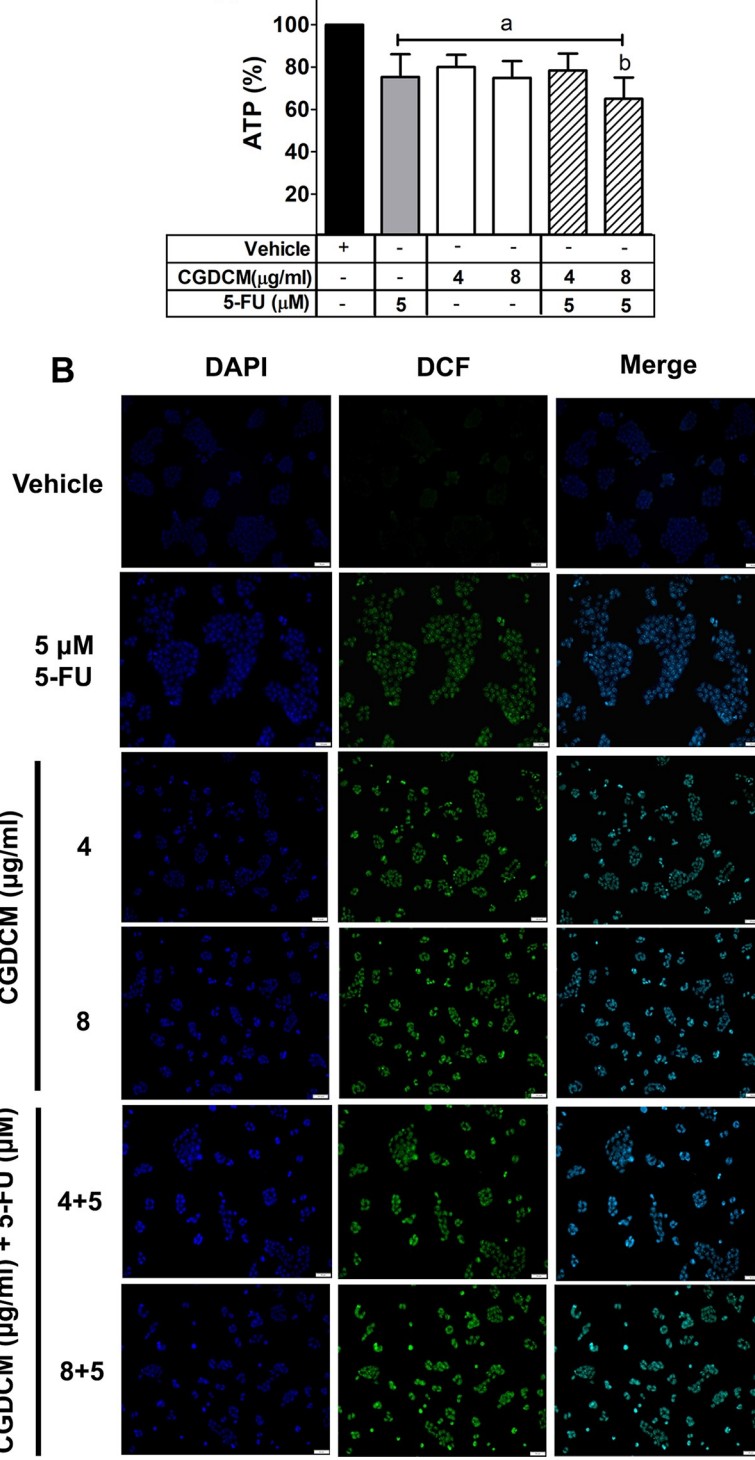

**Fig 6. Effect of CGDCM and 5-FU on the production of ATP and ROS in HCT116 cells.** The levels of ATP and ROS in HCT116 cells after treatment with CGDCM, 5-FU, and combinations of CGDCM (4, 8, and 10 µg/mL) with 5-FU (5 µM or 0.65 µg/mL) for 24 h. The ATP levels in HCT116 cells were analyzed using an ATP assay kit (**A**). The ROS levels in HCT116 cells were analyzed by fluorescence microscopy; bars = 50 µm (**B**). The data are presented as mean ± SD. a; p < 0.05 vs. vehicle group. b; p < 0.05 vs. 5-FU group. Abbreviations: CGDCM, *C. gigantea* dichloromethane extract; 5-FU, 5-fluorouracil; ATP, adenosine triphosphate; DAPI, 4′,6-Diamidino-2-phenylindole dihydrochloride; DCF, 2′,7′-dichlorofluorescein.

green fluorescence of 2′,7′-dichlorofluorescein (DCF) in HCT116 cells increased after treatment with CGDCM (4 and 8 μg/mL), 5-FU (5 μM or 0.65 μg/mL), and their combinations for 24 h (Fig 6B), indicating an increase in ROS levels.

## Discussion

A 95% ethanolic extract of the *C. gigantea* stem barks was prepared with ultrasonication to obtain a crude extract containing various ethanol-soluble compounds. The crude extract was fractionated with DCM, EtOAc, and water to obtain three fractions containing non-polar, semi-polar, and polar compounds, respectively. Quantitative analysis of phytochemicals in the four extracts indicated that all of them contained varying quantities of the five groups of secondary metabolites (Table 1). CGDCM and CGEtOAc were rich in cardiac glycosides, triterpenoids, and phenolic compounds, whereas CGEtOH and CGW contained lower quantities of these classes of phytochemicals. The concentration of flavonoids in the four extracts was in the range of 17.1–47.9 mg RTE/g extract. All four extracts contained only small quantities of alkaloids, ranging from 35.0–112.0 mg BCE/10 g extract. Additionally, calotropin, a cardenolide exhibiting anticancer activity [4,6], was present in the highest concentration in CGEtOH, followed by CGW, CGDCM, and CGEtOAc. These results suggest that CGDCM and CGEtOAc may have a high potential to exhibit strong biological, especially anticancer, activity, owing to the high concentration of diverse chemical components.

In this study, fractions containing more hydrophobic compounds (CGDCM and CGEtOAc) were found to be more cytotoxic toward HCT116 cells, with $IC_{50}$ values of 5.91 and 7.71 μg/mL, respectively. Therefore, CGDCM and CGEtOAc were predicted to contain active ingredients exhibiting a high potential for anticancer activity. Extracts prepared from various parts of the *C. gigantea* plant have been reported to exhibit anticancer activity. The ethanolic extracts prepared from the whole plant exhibited cytotoxicity in A549 and NCI-H1299 non-small cell lung cancer cells after 48 h of treatment [44]. Ethanolic extracts of the roots and leaves of *C. gigantea* have been found to be more cytotoxic toward T47D breast cancer cells after 24 h of treatment, than the extracts prepared from the flowers [15]. Moreover, ethanolic extracts of the leaves of *C. gigantea* have been shown to inhibit the growth of fibrosarcomas through the upregulation of caspase-3 expression (apoptotic index and immunohistochemistry caspase-3 score) in a 7,12 dimethylbenz(α) anthracene acetone-induced mouse model [45]. On the contrary, the cytotoxic effect of the *C. gigantea* leaf extract against colon cancer, WiDr, cells was observed in the DCM, EtOAc, and EtOH fractions, exhibiting $IC_{50}$ values of 40.57, 41.79, and 48.5 μg/mL, respectively [17,46].

This activity has been attributed to the constituent flavonoids and terpenoids. Secondary metabolites present in the plant extract have been reported to exhibit therapeutic efficacy against cancer cells. Flavonoids and terpenoids found in a 70% ethanolic extract of the leaves of *C. gigantea*, as well as its DCM and EtOAc fractions, were found to exhibit cytotoxicity in a human colon cancer, WiDr, cell line [46]. Methanolic and chloroform extracts of the root bark and ethyl acetate extracts of the flowers of *C. gigantea* decreased the viability of tumor cells in Ehrlich ascites carcinoma cell-bearing mice [16]. Several lines of evidence suggest that the cytotoxicity of the *C. gigantea* extracts against cancer cells is attributed to the constituent cardiac glycosides, triterpenoids, and phenolic compounds.

Cardenolides extracted from *C. gigantea* were found to exhibit anticancer activity in triple-negative breast cancers [47], and a root bark extract was reported to exhibit activity in A549 and HeLa cancer cells [6]. The potential cytotoxicity of cardenolides from *C. gigantea* against human breast cancer cells was believed to be mediated by the inhibition of the transcriptional activity of hypoxia-inducible factor-1 (HIF-1) [4]. The eight compounds (uscharin, 15β-

hydroxyuscharin, 19-deoxy-15β-hydroxyuscharin, 2″-oxovoruscharin, calactin, calotropin, gomphoside, and asclepin) isolated from the latex and fruit exhibited cytotoxic effects through the inhibition of HIF-1 in human breast cancer MCF-7 cells [4]. It has been reported that cardenolides (including calactin, calotropagenin, uscharin, afroside, calatoxin, gamphoside, and two unknown ones) in the *C. gigantea* methanolic extract prepared from the leaves and stems inhibited cell growth in breast cancer, MCF-7, cells [31]. Calotropin, a cardenolide isolated from *C. gigantea*, induced apoptotic cell death in SW480 cells by inhibiting the Wnt signaling pathway [48]. Coroglaucigenin extracted from the roots of *C. gigantea* downregulated the expression of cyclin-dependent kinase 4 and the dephosphorylation of Akt to induce autophagy and senescence in CRC cells [49]. Cardenolides, calotropin, calactin, and coroglaucigenin isolated from the EtOAc fraction of an extract prepared from the stems and leaves of *C. gigantea* exhibited anticancer activity in A549 human lung cancer cells [19]. Moreover, calotropin isolated from *C. gigantea* inhibited proliferation in HCT116 and HT-29 cells [50], and three cardenolides, namely *3′-O*-methylcalotropin, *3′-O*-acetylfrugoside, and *3′-O* benzoylfrugoside (1–3), isolated from the bark of Vietnamese *C. gigantea*, exhibited cytotoxicity toward HeLa and A549 cells [18]. Cardenolides isolated from root bark extracts of *C. gigantea* revealed that C-10 formyl and hydroxymethyl groups, including the double-linked six-membered ring sugar unit, enhanced the cytotoxicity of compounds in lung carcinoma, A549, and cervical cancer, HeLa, cells [6].

Triterpenoids have also been reported to act as potential sources of cytotoxicity in cancer cells. Calotroposid A, a triterpenoid glycoside isolated from the ethyl acetate extract of the root of *C. gigantea*, inhibited the growth of WiDr colon cancer cells [51]. Anhydrosophoradial-3-acetate, a triterpenoid isolated from a *C. gigantea* flower EtOAc extract, was found to decrease the cell viability in Ehrlich ascites carcinoma cell-bearing mice [52]. On this basis, it is speculated that cardenolides and triterpenoids may be responsible for the anticancer activity of the *C. gigantea* stem bark extracts in our study.

Although the present study demonstrated the potent cytotoxic effect of CGDCM on HCT116 cells, a combination of low concentrations of CGDCM and 5-FU (5 µM or 0.65 µg/mL) exhibited a stronger cytotoxic apoptotic effect than either of the agents alone. More importantly, this combination exhibited selective cytotoxicity toward cancer cells, which were found to be sensitive to doses that were less toxic to normal fibroblast cells. CGDCM enhanced the anticancer effect of a low dose of 5-FU. Concentrations of 5-FU below 10 µM have been found to exhibit synergistic effects and decrease drug resistance in CRC cells [25]. However, the more a treatment is successful, a higher dose of drugs can be used for cancer cells. High doses of 5-FU are used for long-duration cancer treatments, leading to the development of drug resistance and the incidence of systemic adverse effects [53,54]. Combination therapy using reduced doses of 5-FU and compounds of natural origin exhibiting anticancer activity has been shown to enhance the suppression of cell viability compared to monotherapy with high doses of 5-FU [55]. Combination with plant extracts has also been shown to abolish resistance to 5-FU [56]. Therefore, combination therapy using lower drug doses than those used in monotherapy is an effective therapeutic approach to enhance the anticancer activity, overcome drug resistance, and minimize the undesired side effects.

We also explored the mechanism responsible for the selective cytotoxicity of CGDCM and the CGDCM−5-FU combination in HCT116 cells, which were found to downregulate the production of ATP and upregulate the formation of ROS. ROS accumulation is a cellular stress marker that initiates apoptosis in cancer cells [57]. It was reported that a 95% ethanolic extract prepared from the whole *C. gigantea* plant induced apoptotic cell death through the upregulation of ROS production by suppressing the ROS scavenger superoxide dismutase 2 and catalase in A549 and NCI-H1299 non-small cell lung cancer cells [14]. Coroglaucigenin, a

cardenolide isolated from the EtOAc fraction of an extract prepared from the stems and leaves of *C. gigantea* effectively induced apoptosis in human lung cancer cells through the inhibition of antioxidant molecules (NAD(P)H dehydrogenase [quinone] 1, heme oxygenase 1), leading to the accumulation of ROS [19]. Additionally, bioactive flavonoids, triterpenoids, and phenolics found in a bitter melon extract, prepared from the whole de-seeded fruit, promoted apoptosis through the suppression of the *de novo* lipogenesis pathway and the activation of ROS generation in Cal27 and JHU022 oral cancer cells [58]. Galangin, a flavanol extracted from the roots of *Alpinia galangal*, induced apoptosis in MCF-7 and T47D human breast cancer cells by upregulating the activity of nicotinamide adenine dinucleotide phosphate, leading to an increase in ROS production [59]. Curcumin (diferuloylmethane), a polyphenol isolated from *Curcuma longa* (family: Zingiberaceae), enhanced ROS production and increased the levels of malondialdehyde, a lipid peroxidation marker, resulting in the induction of tumor cell apoptosis in *in vitro* and *in vivo* models [60].

ROS-mediated apoptosis in human bladder cancer cells following treatment with isorhamnetin (3′-methoxy-3,4′,5,7-tetrahydroxyflavone), a flavanol aglycone isolated from the fruit and leaves of various plants, including *Hippophae rhamnoides* L., *Oenanthe javanica*, and *Ginkgo biloba* L., has been reported to involve mitochondrial ATP production. Isorhamnetin has been found to decrease ATP levels, leading to the activation of the mechanistic target of rapamycin (mTOR)/ribosomal protein S6 kinase/acetyl coenzyme A carboxylase 1 signaling pathway, and the induction of apoptosis [61]. ATP is the main source of energy for cellular processes, and a reduction in its generation induces apoptotic cell death in several cancer cells. However, the mechanism of ATP-induced apoptosis of cancer cells triggered by *C. gigantea* extracts has not yet been identified. Wogonoside (wogonin-7-glucuronide, $C_{22}H_{20}O_{11}$), a bioactive flavonoid isolated from the roots of *Scutellaria baicalensis* Georgi, induced apoptosis in human non-small cell lung cancer, A549, cells by promoting mitochondrial dysfunction through the reduction in ATP levels, and the consequent activation of AMP-activated protein kinase/mTOR signaling [62].

A number of reports have suggested that the activation of endoplasmic reticulum (ER) stress is responsible for the potential anticancer activity of the secondary metabolites, including cardiac glycosides, triterpenoids, and phenolic compounds present in *C. gigantea* and other herbal plants. Oleandrin, a cardiac glycoside isolated from the leaves of *Nerium oleander*, induced apoptosis in breast cancer, MCF7 and MDA-MB-231, cells through the activation of ER stress, by enhancing the expression of phospho-protein kinase-like ER kinase, phosphorylating sequential target proteins (eukaryotic translation initiation factor 2α and activating transcription factor 4), and upregulating the expression of the DNA damage-inducible transcript 3 protein [63]. The extract prepared from the whole de-seeded bitter melon fruit is also rich in bioactive compounds, similar to those found in the *C. gigantea* extract. These were found to induce ER stress and induce apoptosis in Cal27 and JHU022 oral cancer cells [58].

The downregulation of pathways involved in fatty acid synthesis is also thought to be associated with the induction of apoptosis by the bioactive compounds present in *C. gigantea*. Kahweol, a coffee-specific diterpene, induced apoptosis in human epidermal growth factor receptor 2-overexpressing cancer cells by suppressing the PI3K/Akt/mTOR/sterol regulatory element-binding transcription factor (SREBP) 1 signaling pathway that regulates fatty acid synthesis [64]. Downregulation of fatty acid synthesis has been found to trigger apoptosis via the Akt/mTOR/SREBP-1c signaling pathway in many cancer cell lines [64–67].

5-FU has long been used for the treatment of various cancer types, including colorectal, breast, and head and neck cancers [21]. *In vivo*, 5-FU is converted to its active metabolites, fluorodeoxyuridine monophosphate, fluorodeoxyuridine triphosphate, and fluorouridine triphosphate, which inhibit the enzyme thymidylate synthase, thus disrupting nucleic acid

synthesis, resulting in cell death [3]. However, 5-FU causes several systemic side effects and is prone to drug resistance [21,53]. 5-FU-resistance in esophageal cancer cells has been reported to be correlated with an upregulation of the PI3K/AKT pathway [68]. Several lines of evidence have demonstrated the potential of compounds derived from plants to enhance the sensitivity of cancer cells to 5-FU. Previous studies have shown that 5-FU applied alone to colon cancer Caco-2 cells caused apoptosis and cell cycle arrest. When used in combination with verbascoside, a phynylethanoid glycoside, cancer cells were further sensitized to 5-FU treatment [25]. In combination with rutin, a quercetin flavonoid glycoside, 5-FU exhibited a synergistic effect in the induction of apoptosis in PC3 prostate cancer cells [24]. Casticin, a polymethexyflavone isolated from *Vitex rotundifolia* L. and other *Vitex* species (family: Verbenaceae), in combination with 5-FU also induced apoptosis in mouse leukemia WEHI-3 cells [69].

In conclusion, in the present study, we show that a combination of 5-FU and the DCM fraction of the *C. gigantea* stem bark extract exhibited enhanced potency in the induction of apoptosis in HCT116 cells, and lower toxicity in normal human fibroblast cells, compared with either CGDCM or 5-FU used alone. Therefore, this extract is a promising alternative anticancer drug, which may be used in combination with base chemotherapeutic regimens to achieve improved safety and efficacy in the treatment of CRC.

## Supporting information

**S1 Fig. The appearance of *Calotropis gigantea*.** (**A**) flowers, (**B**) leaves, (**C**) fruits, (**D**) dry stem bark, and (**E**) the herbarium specimen.
(TIF)

**S2 Fig. HPLC analysis.** (**A**) Chromatogram of calotropin obtained upon HPLC analysis. (**B**) High resolution mass spectrum of calotropin. Chromatograms of CGEtOH (**C**), CGDCM (**D**), CGEtOAc (**E**), and CGW (**F**). Abbreviations: CGEtOH, *C. gigantea* ethanolic extract; CGDCM, *C. gigantea* dichloromethane extract; CGEtOAc, *C. gigantea* ethyl acetate extract; CGW, *C. gigantea* water extract.
(TIF)

**S1 Raw images.**
(PDF)

**S2 Raw images.**
(PDF)

## Acknowledgments

Authors would like to acknowledge Professor Zhi-Hong Jiang and Dr. Li-Ping Bai, Macau University of Science and Technology, Macau, for providing calotropin. We would like to thank Editage (www.editage.com) for English language editing.

## Author Contributions

**Conceptualization:** Dumrongsak Pekthong, Supawadee Parhira, Piyarat Srisawang.

**Data curation:** Supawadee Parhira, Piyarat Srisawang.

**Formal analysis:** Supawadee Parhira, Piyarat Srisawang.

**Funding acquisition:** Supawadee Parhira, Piyarat Srisawang.

**Investigation:** Thanwarat Winitchaikul, Suphunwadee Sawong, Kittiya Kamonlakorn, Supawadee Parhira, Piyarat Srisawang.

**Methodology:** Thanwarat Winitchaikul, Suphunwadee Sawong, Supawadee Parhira, Piyarat Srisawang.

**Project administration:** Dumrongsak Pekthong, Supawadee Parhira, Piyarat Srisawang.

**Resources:** Pranee Nangngam, Supawadee Parhira, Piyarat Srisawang.

**Supervision:** Piyarat Srisawang.

**Validation:** Damratsamon Surangkul, Metawee Srikummool, Julintorn Somran, Dumrongsak Pekthong, Kittiya Kamonlakorn, Pranee Nangngam, Supawadee Parhira, Piyarat Srisawang.

**Visualization:** Dumrongsak Pekthong, Supawadee Parhira, Piyarat Srisawang.

**Writing – original draft:** Thanwarat Winitchaikul, Suphunwadee Sawong, Damratsamon Surangkul, Metawee Srikummool, Julintorn Somran, Dumrongsak Pekthong, Kittiya Kamonlakorn, Pranee Nangngam, Supawadee Parhira, Piyarat Srisawang.

**Writing – review & editing:** Thanwarat Winitchaikul, Suphunwadee Sawong, Damratsamon Surangkul, Metawee Srikummool, Julintorn Somran, Dumrongsak Pekthong, Kittiya Kamonlakorn, Supawadee Parhira, Piyarat Srisawang.

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
