## [Decision Letter · Decision Letter 0]

12 May 2021

PONE-D-21-08509

The Calotropis gigantea Stem Bark Extract Induced Apoptosis Related to Reactive Oxygen Species and Adenosine Triphosphate Production in Colon Cancer Cells

PLOS ONE

Dear Dr. Srisawang,

Thank you for submitting your manuscript to PLOS ONE. After careful consideration, we feel that it has merit but does not fully meet PLOS ONE’s publication criteria as it currently stands. Therefore, we invite you to submit a revised version of the manuscript that addresses the points raised during the review process.

We look forward to receiving your revised manuscript.

Kind regards,

Mukesh Singh Sikarwar

Academic Editor

PLOS ONE

Journal Requirements:

3. During your revisions, please note that a simple title correction is required: to follow correct English language usage, the title should read "Calotropis gigantea stem bark extract induced apoptosis related to ROS and ATP production in colon cancer cells". Please ensure this is updated in the manuscript file and the online submission information.

In your cover letter, please note whether your blot/gel image data are in Supporting Information or posted at a public data repository, provide the repository URL if relevant, and provide specific details as to which raw blot/gel images, if any, are not available. Email us at plosone@plos.org if you have any questions

Additional Editor Comments:

Presented manuscript is on the novel strategies of development of Colon cancer treatment. Combination of CGDCM with 5-FU is the newer approach. Manuscript is recommended for acceptance after minor revision.

Reviewers' comments:

Reviewer's Responses to Questions

**Comments to the Author**

1. Is the manuscript technically sound, and do the data support the conclusions?

Reviewer #1: Yes

Reviewer #2: Yes

Reviewer #3: Yes

2. Has the statistical analysis been performed appropriately and rigorously? 

Reviewer #1: Yes

Reviewer #2: Yes

Reviewer #3: Yes

3. Have the authors made all data underlying the findings in their manuscript fully available?

Reviewer #1: Yes

Reviewer #2: Yes

Reviewer #3: Yes

4. Is the manuscript presented in an intelligible fashion and written in standard English?

Reviewer #1: Yes

Reviewer #2: Yes

Reviewer #3: Yes

5. Review Comments to the Author

Reviewer #1: The manuscript addresses the cytotoxic potentials of the Calotropis gigantea stem bark extract its different fractions. The article is well structured and technically sound. Nevertheless, the authors are recommended to illustrate the method of calotropin quantification for example to add the the equation of the standard calibration curve.

Reviewer #2: The manuscript entitled, “The Calotropis gigantea Stem Bark Extract Induced Apoptosis Related to Reactive Oxygen Species and Adenosine Triphosphate Production in Colon Cancer Cells” is well-organized. The following points are to be addressed.

1. Explain the novelty of the work. Also include the hypothesis of the study in the introduction section.

2. Why the dichloromethane fraction of the C. gigantea stem bark extract exhibited and enhanced

potency with lower toxicity?

3. It is better to calculate the IC50 value and tabulate the data.

4. It is better to plot the same concentration range with similar unit (ug/ml or mM) for extracts and standard drug in the Fig 1.

5. Kindly provide the fig. captions.

6. The differences in the Fig. 2E are not obvious.

Reviewer #3: The manuscript is well written and designated, I recommend it to be accepted in its current form. The experiment have been well conducted. The authors have conducted different analyses which enriched the research.

6. PLOS authors have the option to publish the peer review history of their article (what does this mean?). If published, this will include your full peer review and any attached files.

Reviewer #1: **Yes: **Reham Hassan Mekky

Reviewer #2: **Yes: **Hriday Bera

Reviewer #3: No

---

## [Author Response · Author response to Decision Letter 0]

18 Jun 2021

June 10, 2021

Dr. Mukesh Singh Sikarwar

Academic Editor 

PLOS ONE

Dear Dr. Sikarwar:

We are pleased to have been offered the opportunity to revise and resubmit our manuscript entitled “The Calotropis gigantea Stem Bark Extract Induced Apoptosis Related to Reactive Oxygen Species and Adenosine Triphosphate Production in Colon Cancer Cells” for publication in PLOS ONE. The manuscript ID is PONE-D-21-08509. The revised title, as suggested by you is “Calotropis gigantea stem bark extract induced apoptosis related to ROS and ATP production in colon cancer cells.”

We truly appreciate you and the reviewers for the constructive comments and suggestions on our manuscript. We have carefully considered all the comments while preparing the revised version of the manuscript. We have addressed all the concerns that were raised and hope that the revised manuscript would now be acceptable. Herein, we have detailed the changes made to the manuscript in accordance with the comments. The changes have been tracked in the revised version for the ease of review. We have also scrutinized the entire manuscript for language. We hope that the revision has improved the quality of the manuscript to your satisfaction. 

Thank you for your consideration. We look forward to hearing from you and hope for a positive decision on our manuscript. 

Sincerely,

Piyarat Srisawang,

Supawadee Parhira

E-mail: piyarats@nu.ac.th; supawadeep@nu.ac.th

Response to the Comments from Editor and Reviewers

Editor's comments: 

Presented manuscript is on the novel strategies of development of colon cancer treatment. Combination of CGDCM with 5-FU is the newer approach. Manuscript is recommended for acceptance after minor revision.

Response:

We are thankful to the Editor for the decision to publish our manuscript after minor revision. 

Reviewers’ comments

Reviewer #1: The manuscript addresses the cytotoxic potentials of the Calotropis gigantea stem bark extract its different fractions. The article is well structured and technically sound. Nevertheless, the authors are recommended to illustrate the method of calotropin quantification for example to add the equation of the standard calibration curve.

Response: 

We appreciate the positive feedback from the reviewer. We would like to thank the reviewer for careful and thorough assessment of our manuscript and for providing detailed, valuable, and constructive comments that have helped us improve the quality of our manuscript.

Regarding the method for calotropin quantification, we would like to inform the reviewer that due to the difference in the methods used for quantification of secondary metabolites in C. gigantea stem bark extracts, UV-Vis spectrometry was used to determine the concentrations of cardiac glycosides, triterpenoids, phenolics, flavonoids, and alkaloids, and has been described under subsection “Quantitative analysis of secondary metabolites in C. gigantea stem bark extracts”. Calotropin was used as a standard cardinolide, and its quantitation was standardized by high-performance liquid chromatography (HPLC) as detailed under subsection “HPLC analysis of the C. gigantea stem bark extracts” in the Materials and Methods section (page 7-8).

The equation for the standard calibration curve of calotropin has also been mentioned under subsection “HPLC analysis of the C. gigantea stem bark extracts”. Please refer to the following text added in the revised manuscript:

“The calotropin content in each extract was calculated from the calotropin standard curve (0.2–100 µg/mL, Y = 34377X - 43075, R2 = 0.9991, where Y = peak area at a retention time of 7.43 ± 0.08 min and X = concentration of calotropin (µg/mL)). The results were expressed as milligram calotropin per gram of extract (mean ± standard deviation [SD] of three independent experiments).”

Reviewer #2: The manuscript entitled, “The Calotropis gigantea Stem Bark Extract Induced Apoptosis Related to Reactive Oxygen Species and Adenosine Triphosphate Production in Colon Cancer Cells” is well-organized. The following points are to be addressed.

1. Explain the novelty of the work. Also include the hypothesis of the study in the introduction section.

Response:

We appreciate the positive feedback from the reviewer and are thankful for the valuable suggestion. Accordingly, we have made appropriate modifications in the revised manuscript. We have added the following sentence in the Introduction section:

“However, the mechanism underlying the induction of apoptosis in cancer cells by the stem bark extract of C. gigantea has not yet been evaluated.”

“Therefore, in this study, we aimed to evaluate the effects of C. gigantea stem bark extracts on the inhibition of growth and induction of apoptosis in colon cancer cells. We also evaluated the effects of combinations of C. gigantea stem bark extracts and 5-FU to establish a novel anticancer regimen for future application in cancer therapeutic studies. Moreover, a combination therapy of C. gigantea with the minimum dose of 5-FU was hypothesized to improve the suppression of cancer cell proliferation, compared to the monotherapy. The results of this study may provide a basis for the development of new anticancer strategies involving C. gigantea combination therapy, which may accelerate the treatment outcomes in cancer. There are only a few reports on the phytochemicals present in the C. gigantea stem bark and on their anticancer activities. These findings shed light on the dichloromethane (DCM) fraction of the C. gigantea stem bark extract as a valuable source for purification of new potent anticancer agents in the future.”

2. Why the dichloromethane fraction of the C. gigantea stem bark extract exhibited and enhanced potency with lower toxicity?

Response:

We thank the reviewer for this query. The dichloromethane fractions of the extract of many parts of C. gigantea have been shown to exhibit higher anticancer activity than other fractions in several studies. This is because of several reasons. The first is the presence of compounds with potent anticancer activity, especially cardenolides, in the dichloromethane fraction. As reported previously, uscharin isolated from dichloromethane fraction of leaf extract of C. gigantea had potent anticancer activity (Jacinto et al., 2011). You et al. (2013) isolated 16 cardenolide compounds from root bark extracts of C. gigantean, which showed potent cytotoxicity in many cancer cell lines.

C. gigantea is rich in cardiac glycosides. A number of cardenolides, such as uscharin, calactin, calotropin, and calotoxin, have been isolated from this plant and shown to exhibit anticancer activity. The total cardiac glycoside content in each of the extracts is summarized in Table 1. CGDCM had the high content of total cardiac glycosides and other secondary metabolites, and calotropin. Thus, we hypothesized that cardenolides being rich in CGDCM is the major compound contributing the most to the anticancer effect in HCT116 and HT-29 cells.

In addition, the higher cytotoxic potency of CGDCM than of other fractions is because of the difference in the responsiveness of cancer cells to the different fractions. Different extracts have different cytotoxic effects on different cell lines. In this study, CGDCM and CGEtOAc fractions contained the same content of active ingredients but CGDCM was found to be slightly more cytotoxic towards HCT116 cells than CGEtOAc.

Regarding the C. gigantea stem bark extract exhibiting lower toxicity in normal cells, we speculate that the mechanism of cytotoxic effect of CGDCM fraction is specific for cancer cells but not for normal cells. We found that CGDCM downregulated the production of ATP and upregulated the formation of ROS. Normally, the intracellular redox status is regulated by the balance of the ratio of superoxide to H2O2, and this ratio is controlled by the constitutive expression of antioxidant enzymes, the SODs. Cancer cells are deficient in antioxidant enzymes. This leads to accumulation of superoxide to induce apoptosis in cancer cells. ROS also suppresses activated Akt in cancer cells, leading to sensitization of the cells to oxidative stress-induced apoptosis (Han et al., 2009; Sun et al., 2014; Moreira et al., 2017). Our results are consistent with those reported previously. The extracts from C. gigantea had opposite effects on cancer and normal cells. The extract from C. gigantea suppressed the viability of lung cancer cells, but not of human normal keratinocytes. The extract enhanced ROS generation and reduced the expression of ROS scavengers (Lee et al., 2019). Likewise, the extract from C. gigantea decreased radiation-induced antioxidant molecules in lung cancer cells but increased their levels in the normal lung cells (Sun et al., 2017). Altogether, we suggested that enhanced ROS generation following treatment with the C. gigantea stem bark extract selectively caused apoptosis in cancer cells.

In addition to ROS, the selective cytotoxicity of CGDCM involved downregulation of ATP production. Extracellular or intercellular ATP concentration is reported to be significantly increased in tumors but is very low in normal healthy tissues. Warburg effect has been identified as the hallmark of cancer cells, which have high demand for ATP synthesis from glucose, amino acids, and fatty acids. Glucose is used for glycolytic ATP production, followed by lactate production, rather than through the TCA cycle regardless of the O2 level. It was found that mitochondrial activities in cancer cells are reduced due to hypoxia, which is related to increased generation of ROS. In contrast to normal cells, de novo fatty acid synthesis occurs at higher rates in tumor tissues to convert excessive dietary glucose to storage lipids (triglycerides) in cells. It was previously reported that inhibition of fatty acid synthesis selective killed tumor cells without affecting the normal tissues. This was followed by a decrease in ATP production (Wang et al., 2017). We speculated that the extract from C. gigantea selectively targeted the de novo fatty acid synthesis that suppressed ATP production in cancer cells without affecting the normal cells.

In addition, the mechanism of inhibition of ATP production by the C. gigantea extract may involve activated insulin-like growth factor-1 receptor (IGF-1/IGF-1R) signal pathway that upregulates the downstream phosphatidylinositol-3-kinase (PI3K/Akt/mTOR) pathway, wherein ATP could be produced in large amounts and provide sustained energy supply for tumor cells to survive. Hu et al. (2020) reported hyperactivation of the IGF-1/IGF-1R-PI3K/Akt/mTOR signal pathway in colon cancer tissues comparing with that in the normal colon tissue. The study showed that Arca subcrenata Lischke, the marine invertebrate species, significantly suppressed the growth of HT-29 cells through the IGF-1/IGF-1R-PI3K/Akt/mTOR signal pathway with little effect on the viability of rat jejunal epithelium, IEC-6, cells and normal colonic mucosa of mice. Thus, selective apoptosis induction in cancer cells by the extract of C. gigantea involved inhibition of ATP production, which might be mediated by the IGF-1/IGF-1R-PI3K/Akt/mTOR signal pathway.

3. It is better to calculate the IC50 value and tabulate the data.

Response:

We thank the reviewer for this suggestion. We have added Table 2 wherein we have presented the IC50 values of fractions from C. gigantea stem bark extracts on HCT116 and HT-29 cells at 24 h. The IC50 values of 5-FU have been expressed in �g/mL and �M in this table and throughout the manuscript.

Table 2. IC50 Values of Fractions from Calotropis gigantea Stem Bark Extracts on HCT116 and HT-29 Cells at 24 h of Treatment.

Extracts IC50 (µg/mL)

 HCT116 HT-29

CGEtOH 32.8 ± 0.83 60.6 ± 3.61

CGDCM 5.9 ± 0.62 44.0 ± 4.06

CGEtOAc 7.7 ± 0.89 43.6 ± 9.34

CGW 39.0 ± 8.88 86.7 ± 10.81

5-FU 34.1 ± 3.57

(248 ± 29.62 µM) 457.1 ± 29.63

(3,598 ± 199.83 µM)

Abbreviations: CGEtOH, C. gigantea ethanolic extract; CGDCM, C. gigantea dichloromethane extract; CGEtOAc, C. gigantea ethyl acetate extract; CGW, C. gigantea water extract; 5-fluorouracil, 5-FU.

4. It is better to plot the same concentration range with similar unit (ug/ml or mM) for extracts and standard drug in the Fig 1.

Response:

We would like to than the reviewer for this suggestion. Many researches have reported the unit of the crude and fractionated extracts in mg/mL or µg/mL owing to the lack of information regarding the content of compounds present in the extracts, whereas the standard or positive control drugs are reported in mg/mL or µg/mL or �M.

Herein, we have shown data in a similar unit, µg/mL. For Fig 1C and 1D, we thought that data for 5-FU expressed in �M appears more understandable. Thus, we surmised that expressing the values with different units did not affect the pattern of responsiveness of cancer cells to the different compounds. We compared the percentage of the cytotoxic effect of the extracts to that of the standard drug. The IC50 value of the standard was used for the combination treatment. However, for the benefit of readers, the unit for IC50 of 5-FU has been expressed in �g/mL as well as �M throughout the manuscript.

Fig 1. Cytotoxic Effects of Calotropis gigantea Stem Bark Extracts on HCT116 and HT-29 Cells. The viability of HCT116 (A) and HT-29 cells (B), evaluated by MTT assay. The vehicle control group was 0.8% DMSO. The effect of the positive control, 5-FU, on HCT116 (C) and HT-29 (D) cells. The data are presented as mean ± SD from a minimum of three independent experiments, and were analyzed using one-way ANOVA with Tukey’s HSD test. *p < 0.05 compared with the vehicle. Abbreviations: CGEtOH, C. gigantea ethanolic extract; CGDCM, C. gigantea dichloromethane extract; CGEtOAc, C. gigantea ethyl acetate extract; CGW, C. gigantea water extract; 5-FU, 5-fluorouracil.

5. Kindly provide the fig. captions.

Response:

We would like to submit to the reviewer that in accordance with the PLOS ONE submission guidelines, figure captions are included in the manuscript text following the paragraph in which the figure is first cited. As such, the captions are not part of the figure files and have not been submitted in a separate document.

6. The differences in the Fig. 2E are not obvious.

Response:

Thank you for your concern. Results of the wound healing assay were presented in Fig 2E and calculated as percentage of the gap distance in Fig 2F. The gap distance for the vehicle decreased in a time-dependent manner, suggesting normal cell proliferation. In contrast, for the treatments, the gap distance remained unchanged after treatments for 12, 24, and 48 h, suggesting inhibition of cell proliferation and migration. 

Reviewer #3: The manuscript is well written and designated, I recommend it to be accepted in its current form. The experiment has been well conducted. The authors have conducted different analyses which enriched the research.

Response:

We appreciate the positive feedback from reviewer #3. We would like to thank the reviewer for investing time and effort in reviewing our manuscript.

---

## [Decision Letter · Decision Letter 1]

28 Jun 2021

Calotropis gigantea stem bark extract induced apoptosis related to ROS and ATP production in colon cancer cells

PONE-D-21-08509R1

Dear Dr. Srisawang,

We’re pleased to inform you that your manuscript has been judged scientifically suitable for publication and will be formally accepted for publication once it meets all outstanding technical requirements.

Kind regards,

Mukesh Singh Sikarwar

Academic Editor

PLOS ONE

Additional Editor Comments (optional):

Reviewers' comments:

Reviewer's Responses to Questions

**Comments to the Author**

1. If the authors have adequately addressed your comments raised in a previous round of review and you feel that this manuscript is now acceptable for publication, you may indicate that here to bypass the “Comments to the Author” section, enter your conflict of interest statement in the “Confidential to Editor” section, and submit your "Accept" recommendation.

Reviewer #1: All comments have been addressed

Reviewer #2: All comments have been addressed

2. Is the manuscript technically sound, and do the data support the conclusions?

Reviewer #1: Yes

Reviewer #2: Yes

3. Has the statistical analysis been performed appropriately and rigorously? 

Reviewer #1: Yes

Reviewer #2: Yes

4. Have the authors made all data underlying the findings in their manuscript fully available?

Reviewer #1: Yes

Reviewer #2: Yes

5. Is the manuscript presented in an intelligible fashion and written in standard English?

Reviewer #1: Yes

Reviewer #2: Yes

6. Review Comments to the Author

Reviewer #1: Dear Authors,

Thanks very much for addressing all the comments and the concerns. I think the manuscript is ready for publication.

Reviewer #2: The various points of manuscript entitled, “The Calotropis gigantea Stem Bark Extract

Induced Apoptosis Related to Reactive Oxygen Species and Adenosine Triphosphate

Production in Colon Cancer Cells” are well

addressed

7. PLOS authors have the option to publish the peer review history of their article (what does this mean?). If published, this will include your full peer review and any attached files.

Reviewer #1: **Yes: **Reham Hassan Mekky

Reviewer #2: No

---

## [Editor Report · Acceptance letter]

23 Jul 2021

PONE-D-21-08509R1 

*Calotropis gigantea* stem bark extract induced apoptosis related to ROS and ATP production in colon cancer cells 

Dear Dr. Srisawang:

I'm pleased to inform you that your manuscript has been deemed suitable for publication in PLOS ONE. Congratulations! Your manuscript is now with our production department. 

Kind regards, 

on behalf of

Dr. Mukesh Singh Sikarwar 

Academic Editor

PLOS ONE